# Humanoid Locomotion as Next Token Prediction

**Ilija Radosavovic**
UC Berkeley

**Bike Zhang**[*]
UC Berkeley

**Baifeng Shi**[*]
UC Berkeley

**Jathushan Rajasegaran**[*]
UC Berkeley

**Sarthak Kamat**
UC Berkeley

**Trevor Darrell**[†]
UC Berkeley

**Koushil Sreenath**[†]
UC Berkeley

**Jitendra Malik**
UC Berkeley

## Abstract

We cast real-world humanoid control as a next token prediction problem, akin to predicting the next word in language. Our model is a causal transformer trained via autoregressive prediction of sensorimotor sequences. To account for the multi-modal nature of the data, we perform prediction in a modality-aligned way, and for each input token predict the next token from the same modality. This general formulation enables us to leverage data with missing modalities, such as videos without actions. We train our model on a dataset of sequences from a prior neural network policy, a model-based controller, motion capture, and YouTube videos of humans. We show that our model enables a real humanoid robot to walk in San Francisco zero-shot. Our model can transfer to the real world even when trained on only 27 hours of walking data, and can generalize to commands not seen during training. These findings suggest a promising path toward learning challenging real-world control tasks by generative modeling of sensorimotor sequences.

## 1 Introduction

The last decade of artificial intelligence (AI) has shown that large neural networks trained on diverse datasets from the Internet can lead to impressive results across different settings. The core enablers of this wave of AI have been large transformer models [37] trained by generative modeling of massive quantities of language data from the Internet [26, 6, 27, 4]. By predicting the next word, these models acquire rich representations of language that can be transferred to downstream tasks [26], perform multi-task learning [27, 25], and learn in a few-shot manner [4].

Are such modeling techniques exclusive to language? Can we learn powerful models of sensory and motor representations in a similar fashion? Indeed, we have seen that we can learn good representations of high-dimensional visual data by autoregressive modeling [5] and related masked modeling approaches [11]. While there has been positive signal on learning sensorimotor representations in the context of manipulation [28], this area remains largely unexplored.

In this paper, we cast humanoid control as data modeling of large collections of sensorimotor sequences. Like in language, we train a general transformer model to autoregressively predict the sequences. In contrast to language, the nature of data in robotics is different. It is high-dimensional and contains multiple input modalities. Different modalities include sensors, like joint encoders or inertial measurement units, as well as motor commands. These give rise to *sensorimotor sequences* which we view as the *sentences* of the physical world. Adopting this perspective suggests a simple instantiation of the language modeling framework in the robotics context. We tokenize the input sequences and train a causal transformer model to predict future tokens. Importantly, we predict *complete* input sequences, including both sensory and motor tokens. In other words, we are modeling the *joint* data distribution as opposed to the conditional action distribution.

---

[*],[†]Authors with stars and daggers listed in arbitrary order. Author contributions listed at the end of the paper.

38th Conference on Neural Information Processing Systems (NeurIPS 2024).

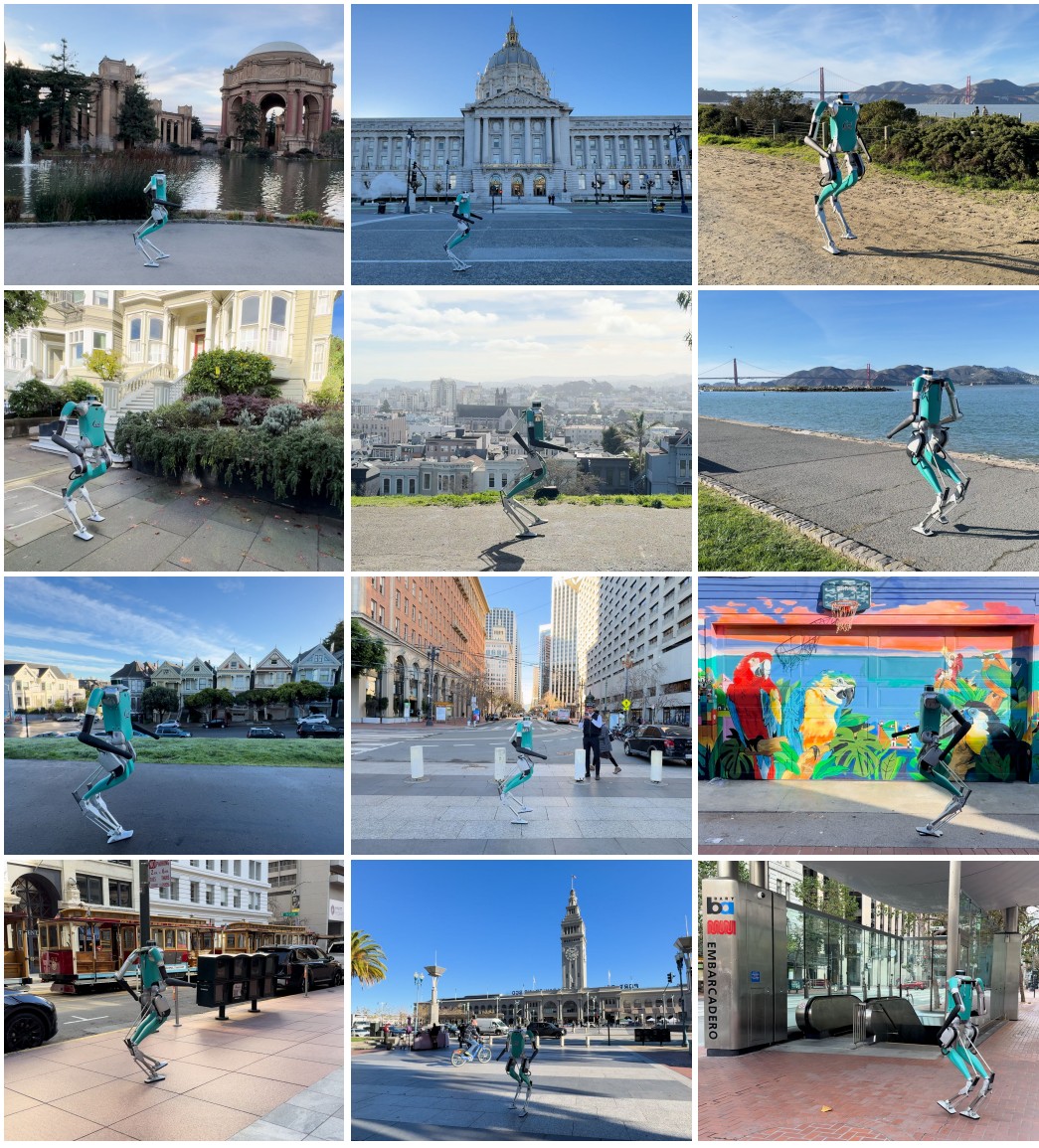

Figure 1: **A humanoid that walks in San Francisco.** We deploy our model to the robot and test it in various locations in San Francisco over the course of one week. Please see our project page for videos. We show that our model enables the humanoid walk over different surfaces including walkways, concrete, asphalt, tiled plazas, and sanded roads. Our model can follow omnidirectional velocity commands well and enables deployment in a challenging city environment like San Francisco.

Our core observation is that if a sequence is incomplete, *i.e.*, some of the sensory or motor information is missing, we can still learn from it by predicting whatever information is present and replacing the missing tokens with learnable mask tokens. The intuition is that if the model has learned to make good predictions, even in the absence of information, it will have acquired a better model of the physical world. An important source of such data are human videos from the Internet. Namely, we can observe human movement in videos but we do not get access to the motor commands or complete sensory inputs. We demonstrate that our method can learn from such data sources effectively.

To validate our method, we apply it to the challenging problem of real-world humanoid locomotion using a full-sized humanoid robot. We first construct a dataset of sensorimotor sequences in simulation. These include complete sequences from a neural network policy trained with reinforcement learning [29], as well as incomplete sequences from three different sources: (i) a model-based controller, (ii) motion capture of humans, and (iii) videos of humans. We reconstruct human videos using computer vision techniques and approximately retarget both the motion capture and video sequences via inverse kinematics. We then train an autoregressive transformer to model this dataset. At test time, we execute the predicted motor commands and ignore the sensory predictions.

We demonstrate that our model can be deployed in the real world zero-shot. Specifically, we deploy our model to a range of different locations and surfaces in San Francisco over the course of one week. Please see Figure 1 for example images and project page for videos. To quantitatively evaluate different aspects of our approach, we perform experiments in simulation. We find that our models trained via sequence modeling alone can be comparable to the state-of-the-art reinforcement learning approach [29] in the settings where data is available. We further find that our approach can benefit from incomplete data, has favorable scaling properties, and can generalize to unseen commands. We encourage the readers to see the arXiv version of this work for additional experiments.

These findings suggest a promising path toward learning challenging real-world robotic control tasks by generative modeling of large collections of sensorimotor sequences.

## 2 Related Work

**Generative models.** Generative models have been studied extensively, starting from Shannon's work [33] to the modern era of large language models [4]. Various such models emerged over the last decade. Examples include GAN [10] and Diffusion models [35, 14] for generating pixels, LSTM [15] and GPT [26] for generating language tokens. These models have been adopted for other modalities as well [23, 9, 38]. Among these, autoregressive transformer models became the front runner, due to the impressive scaling behaviors [17] and ability to learn from in-context examples [3]. They have been successfully extended to other modalities as well, such as vision [5] and vision-language [32]. We explore autoregressive generative models in the context of real-world humanoid locomotion.

**Transformers in robotics.** Following the success of transformer models [37] in natural language processing [26, 6, 27, 3] and computer vision [7, 11], over the last few years, there has been an increased interest in using transformer models in robotics. We have seen a number of works showing that transformers can be effective with behavior cloning. For example, [34] learns multi-task transformer policies with language, [2] trains language-conditioned manipulation policies from large-scale data, [8] trains language models with embodied data, and [1] trains policies conditioned on image goals. We have also seen that transformer models can be effective for large-scale reinforcement learning [29]. Related to ours, [28] learns sensorimotor representations with masked prediction. Like this body of work, we share the goal of using transformer models for robotics but focus on autoregressive modeling of sensorimotor sequences in the context of humanoid locomotion.

**Humanoid locomotion.** Legged locomotion has been a long-standing challenge in robotics. Humanoid locomotion is particularly challenging and has been studied extensively over the the past several decades [18, 13]. Stable locomotion behaviors have been demonstrated through model-based control approaches [30, 16], while optimization-based methods can further enable highly dynamic humanoid motions [19]. Although significant progress has been made, learning-based approaches have the potential to facilitate future progress due to their potential to improve from data and generalize to new environments. Recently, we have seen that a purely learning based approach trained with large-scale reinforcement learning in simulation can enable real-world humanoid locomotion [29]. We use the same architecture, an autoregressive transformer, but propose a different training procedure. Specifically, we train the model with sequence modeling instead of reinforcement learning.

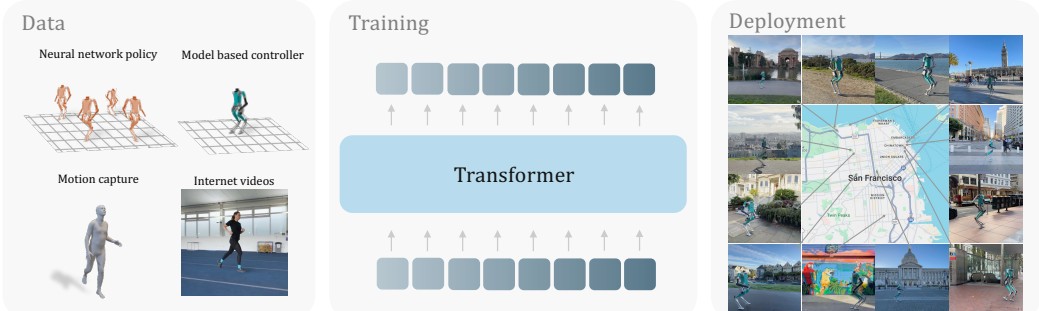

Figure 2: **Humanoid locomotion as next token prediction.** We collect a dataset on trajectories from various sources, such as from neural network policies, model-based controllers, human motion capture, and YouTube videos of humans. Then we use this dataset to train a transformer policy by autoregressive modeling of observations and actions. Our transformer allows a humanoid to walk zero-shot on various terrains around San Francisco. Please see our project page for video results.

## 3 Approach

In this section, we assume a dataset $\mathcal{D}$ of sensorimotor trajectories $\mathcal{T}$ and describe our approach.

### 3.1 Objective

Each sensorimotor trajectory is a sequence of observations, such as joint positions, and actions: $\mathcal{T} = (o_1, a_1, o_2, a_2, ..., o_T, a_T)$. We first tokenize the sequence into K tokens to obtain $t = (t_1, t_2, ..., t_K)$. Our goal is to train a neural network to model the density function $p(t)$ autoregressively:

$$p(t) = \prod_{k=1}^{K} p(t_k | t_{k-1}, ..., t_1) \tag{1}$$

We train our model by minimizing the negative log-likelihood over our trajectory dataset, assuming a Gaussian distribution with constant variance:

$$L = \sum_{t \in \mathcal{D}} - \log p(t) \tag{2}$$

Instead of regressing the continuous values of token elements, we could quantize each dimension into bins or perform vector quantization. However, we found the regression approach to work reasonably well in practice and opt for it in our experiments for simplicity.

### 3.2 Missing modalities

In the discussion so far we have assumed that each trajectory is a sequence of observations and actions. Next, we show how our framework can be generalized to sequences with missing modalities, like trajectories extracted from human videos that do not have actions. Suppose we are given a trajectory of observations without the actions $\mathcal{T} = (o_1, o_2, ..., o_T)$. Our core observation is that we can treat a trajectory without actions like a regular trajectory with actions masked. Namely, we can insert mask tokens [M] to obtain $\mathcal{T} = (o_1, [M], o_2, [M], ..., o_T, [M])$. This trajectory now has the same format as the regular trajectories and can thus be processed in a unified way. We ignore the loss for the predictions that correspond to the masked part of inputs. Note that this principle is not limited to actions and applies to any other modality as well, such as partially missing sensory observations.

### 3.3 Aligned prediction

Rather than predicting the next token in a modality-agnostic way, we make predictions in a modality-aligned way. Namely, instead of predicting the next token in the sequence, for each input token we predict the next token of the *same* modality. Please see Figure 3 for diagrams.

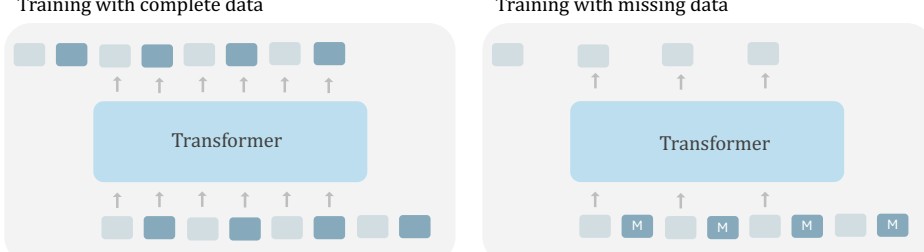

Figure 3: **A general framework for training with different data sources.** Our data modeling allows us to train our transformer with multiple modes of training. In the case of observation-action pairs being available, we train our transformer to predict the next pair of observation-action. When there is no action data available, with MoCap and internet data, we only train our transformer to predict the next observations by masking the actions with a mask token. These two models of training allow our model to utilize both types of data, and this enables us to scale our training in terms of data.

## 3.4    Joint training

We have two options for training on collections that contain diverse trajectories in terms of noise levels or modality subsets. We can either train jointly with all data at once, including complete and incomplete trajectories. Alternatively, we can first pre-train on noisy and incomplete trajectories. This can be viewed as providing a good initialization for then training on complete trajectories. And then fine-tune the model on complete data with actions. We find that both approaches work comparably in our setting and opt for joint training in the majority of the experiments for simplicity.

## 3.5    Model architecture

Our model is a transformer [37] with a fairly standard architecture. Given the trajectories from either complete or incomplete data, we first tokenize the trajectories into tokens. We learn separate linear projection layers for each modality but share weights across time. To encode the temporal information we use positional embeddings. Let us assume $o_i \in \mathcal{R}^m$ and $a_i \in \mathcal{R}^n$, then:

$$t_i = \texttt{concat}(o_i, a_i), \tag{3}$$

$$h_i^0 = W t_i, \tag{4}$$

where $W \in \mathcal{R}^{d \times (m+n)}$ is a linear projection layer to project concatenated observation and action modalities to $d$ dimensional embedding vector. The superscript 0 indicates the embedding at 0-th layer, *i.e.*, the input layer. When action is unavailable, we use a mask token $[\texttt{M}] \in \mathcal{R}^n$ to replace $a_i$, and $[\texttt{M}]$ is initialized as a random vector and learned end-to-end with the whole model. The model takes the sequence of embedding vectors $H_0 = \{h_1^0, h_2^0, ..., h_t^0\}$ as input.

The transformer architecture contains $L$ layers, each consisting of a multi-head self-attention module and an MLP module. Assume the output of the layer $l$ is $H_l$, then the layer $l+1$ output is computed as follows:

$$\tilde{H}_l = \texttt{LayerNorm}(H_l) \tag{5}$$

$$\tilde{H}_l = \tilde{H}_l + MHSA(\tilde{H}_l) \tag{6}$$

$$H_{l+1} = \tilde{H}_l + MLP(\tilde{H}_l) \tag{7}$$

Here, the multi-head self-attention has causal masking, where the token only attends to itself and the past tokens. Once the tokens are processed through all the layers, we project the embedding to predicted states and actions, by learning a linear projection layer $\widehat{W} \in \mathcal{R}^{(m+n) \times d}$:

$$\widehat{t}_{i+1} = \widehat{W} h_i^L \tag{8}$$

$$\widehat{o}_{i+1} = (\widehat{t}_{i+1})_{0:m} \tag{9}$$

$$\widehat{a}_{i+1} = (\widehat{t}_{i+1})_{m:(m+n)} \tag{10}$$

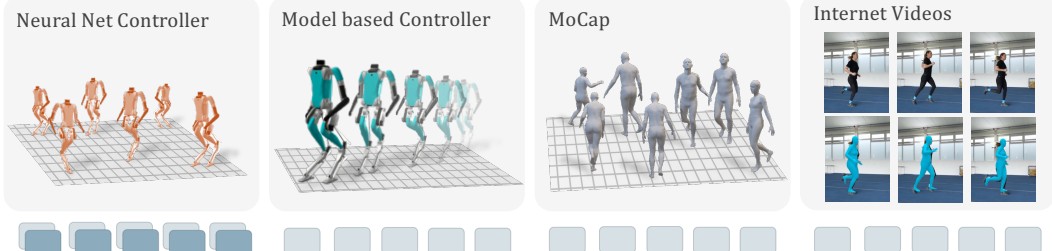

Figure 4: **Training dataset.** To train our model, we construct a dataset of trajectories coming from four different sources. *(i) neural network policy:* provides trajectories with complete observations and actions. *(ii) model-based controller:* produces trajectories from the same robot morphology but without actions. *(iii) motion capture of humans:* does not contain actions and is approximately retargeted onto the robot. *(iv) internet videos of humans:* sequences of human poses are first reconstructed via computer vision techniques and then approximately retargeted onto the robot.

Then we train the transformer with the objective in (2). In the cases where the token is masked, we do not apply any losses. We train our transformer with both types of data, as shown in Figure 3. This allows us to use various sources of data, thus enabling scaling in terms of data.

### 3.6 Model inference

At inference time, our transformer model always has access to observation-action pairs. In this setting, we apply our transformer model autoregressively for each observation-action pair token. By conditioning on past observations and actions, we predict the next action (or observation-action pair) and execute the predicted action. We then record the ground-truth observation from the robot and discard the predicted observation. We use the ground-truth observation and predicted action as the next set of tokens and append them to the past tokens to predict the next observation-action pair.

## 4 Dataset

Our approach motivates building a dataset of trajectories for training our model. We construct a dataset of trajectories from four different sources: (i) a neural network policy, (ii) a model-based controller, (iii) human motion capture, and (iv) human videos from YouTube. An illustration of trajectories from different data sources is shown in Figure 4. We describe each data source next.

### 4.1 Neural network trajectories

As the first source of training trajectories, we use a state-of-the-art neural network policy trained with large-scale reinforcement learning in simulation [29]. Specifically, this policy was trained with billions of samples from thousands of randomized environments in Isaac Gym [22]. We run this policy in the physics simulator developed by Agility Robotics and collect 10k trajectories of 10s each on flat ground, without domain randomization. Each trajectory is conditioned on a velocity command sampled from a clipped normal distribution as follows: linear velocity forward $[0.0, 1.0]$ m/s, linear velocity sideways $[-0.5, 0.5]$ m/s, and turning angular velocity $[-0.5, 0.5]$ rad/s.

Since we have access to the data generation policy, we are able to record complete observations as well as the exact actions that the model predicted. We use this set as our source of complete sensorimotor trajectories that have complete observations as well as ground truth actions.

### 4.2 Model-based trajectories

As the second source of trajectories, we use the model-based controller developed by Agility Robotics. It is the controller that is deployed on the Digit humanoid robot and available in the simulator provided by Agility Robotics as well. We collect 20k trajectories of walking on a flat ground of 10 seconds each, where we sample the velocity commands as follows: linear velocity forward $[-1.0, 1.0]$ m/s, linear velocity sideways $[-1.0, 1.0]$ m/s, and turning angular velocity $[-1.0, 1.0]$ rad/s.

The model-based controller outputs joint torques, which are not consistent with our joint position action space. Thus, we only record the observations without the actions. This data serves as a source of trajectories with accurate observations from the same robot morphology but without the actions.

### 4.3 Human motion capture trajectories

As the next source of trajectories, we use the motion capture (MoCap) recordings of humans from the KIT dataset [24] distributed via the AMASS repository [21]. This data was recorded using optical marker-based tracking in a laboratory setting. The dataset consists of ~4k trajectories. We use a subset of ~1k standing and walking trajectories. We exclude motions like dancing and jumping.

In addition to not containing the ground truth actions, the MoCap trajectories come with an additional challenge: different morphology. Namely, MoCap trajectories capture *human* keypoint positions in 3D and over time. In order to use these trajectories for training a robot, we solve an inverse kinematics problem to approximate the corresponding robot poses. Please see the arXiv for additional details.

### 4.4 Trajectories from YouTube videos

Internet videos of people doing various activities are potentially a vast source of training data for humanoid robots. However, the raw pixels have no information about the state and actions of the human. To recover this, we first we run a computer vision tracking algorithm PHALP [31] to extract sequences of 3D human poses. This provides an estimate of the 3D joints of the human body SMPL [20] parameters and a noisy estimate of the human joints in the world coordinates.

We use the human body joint positions to retarget the motion to the humanoid robot using the inverse kinematics optimization, like in the case of motion capture data discussed previously. After we retarget the motions from human videos to humanoid robot trajectories, we filter out the trajectories with high reconstruction error. Note that the scale of this data comes with the cost of being noisy.

## 5 Experiments

We evaluate our approach on the challenging task of humanoid locomotion. We perform outdoor experiments on real hardware and systematic evaluations in simulation. We encourage the readers to see the extended version of this work on arXiv, which includes additional experiments and ablations.

### 5.1 Experimental setup

**Robot platform.** We use the Digit humanoid robot developed by Agility Robotics. It is a full-sized humanoid that is 1.6m tall and weighs 45 kilograms. The robot has 36 degrees of freedom including the floating base, 20 of which are actuated. Due to its high dimensionality and the four-bar linkage structure, it is challenging to simulate accurately which makes it particularly interesting for learning approaches like ours that can learn from data, without requiring an explicit model or a simulator.

**Model.** Our model has a hidden size of 192 dimensions, with 4 layers of self-attention layers and MLP layers. Each self-attention has 4 heads. We use LayerNorm before each attention layer and ReLU activation after the MLP layer. We use a BatchNorm layer to process the input before the transformer model. When predicting a token at time $k$, to keep the context length at a reasonable size, we only keep the past 16 steps in input. In Section 5.6, we show the model is able to scale up to more parameters and longer context length and achieve higher performance. We train on 4 NVIDIA A100s.

### 5.2 Real-world deployment

We begin by reporting the results from the real-world experiments. Specifically, we deploy our model to the robot and evaluate it at various locations in San Francisco over the course of one week. Please see Figure 1 for examples and project page for videos. We find that our model is able to walk over a variety of surfaces including walkways, concrete, asphalt, tiled plazas, and dirt roads. Note that the deployment in a large city environment, like San Francisco, is more challenging than in constrained laboratory environments. The city environment is crowded with pedestrians, less controlled, and less forgiving. This makes the error tolerance low and requires a model that works consistently well.

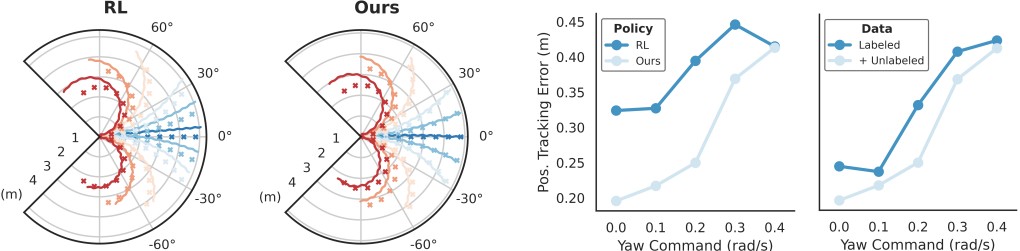

Figure 5: **Comparison to state of the art, trajectory adherence.** The robot is commanded to walk starting from the origin with a fixed heading command of 0.5 m/s and varying yaw commands in $[-0.4, 0.4]$ rad/s. We plot the desired (dotted) and actual (solid) trajectories for our policy and a reinforcement-learning trained policy (RL).

Figure 6: **Tracking error comparisons.** *Left:* We find that our model can follow yaw commands more accurately than the state-of-the-art RL policy [29]. *Right:* We see that our model can benefit from unlabeled trajectories without actions which is a promising signal for scaling our method to large datasets of diverse trajectories.

## 5.3 Evaluation metrics

Next, we evaluate the models in simulation under two metrics: *tracking error* and *prediction error*. Tracking error measures how accurately the robot follows a specific command. The prediction error is the next token prediction loss measured on a separate set of validation data. We introduce two metrics with details as follows and show that two metrics can consistently predict locomotion performance.

**Tracking error.** In all experiments, the robot starts from rest in a simulated environment and is issued a constant natural walking command consisting of a desired heading velocity sampled in $[0.35, 0.70]$ m/s, angular velocity sampled in $[-0.4, 0.4]$ rad/s, and zero lateral velocity. We compute $\mathbf{x}^*(t)$, the ideal robot base position trajectory that fully satisfies the velocity command $\mathbf{v}^*(t)$ at all time steps. To measure the accuracy of command tracking, we define the position tracking error as $\frac{1}{T} \sum_{t=0}^{T} \|\mathbf{x}(t) - \mathbf{x}^*(t)\|$. Each trajectory lasts for a duration of 10 seconds. For all evaluation experiments, we use the MuJoCo simulator [36] which is able to simulate the Digit robot accurately.

**Prediction error.** Since the model is trained with the next token prediction, we evaluate the prediction error on a set of validation data that is held out from training data and contains state-action trajectories collected from the RL policy. This is similar to the language modeling evaluation for large language models [12]. We test both state and action prediction errors and add them together as the error metric.

## 5.4 Comparison to the state of the art

**Trajectory adherence.** We compare our model to a neural network controller trained with reinforcement learning (RL) [29]. Figure 5 presents a visual comparison of the trajectory adherence of our controller against these state-of-the-art baselines. Starting with a robot at the origin, we plot the actual trajectory of the robot with eleven different yaw commands selected from $\{0.00, \pm 0.05, \pm 0.10, \pm 0.20, \pm 0.30, \pm 0.40\}$ rad/s. For each model, we jointly plot the desired and actual path traced by the robot base. Our model exhibits better tracking than the RL controller at all turning speeds, and achieves close to perfect tracking for straight-line walking.

**Quantitative evaluation.** In Figure 6, left, we repeat the above comparison to the RL controller ($N = 245$), with the full range of heading and yaw velocities mentioned in Section 5.3. We plot the mean position tracking error, binned by the commanded angular yaw. While both models have lower tracking errors at lower yaw, ours consistently outperforms the baseline RL policy. Note that our model was trained on a dataset that includes the trajectories from the same baseline RL policy.

## 5.5 Training with action-free data

One of the benefits of our approach is that it can be applied to trajectories from diverse sources, including with missing information such as actions, like in the case of human videos. In Figure 6, right, we compare the performance of training only with complete trajectories to joint training with both complete and incomplete trajectories. We observe that including incomplete trajectories

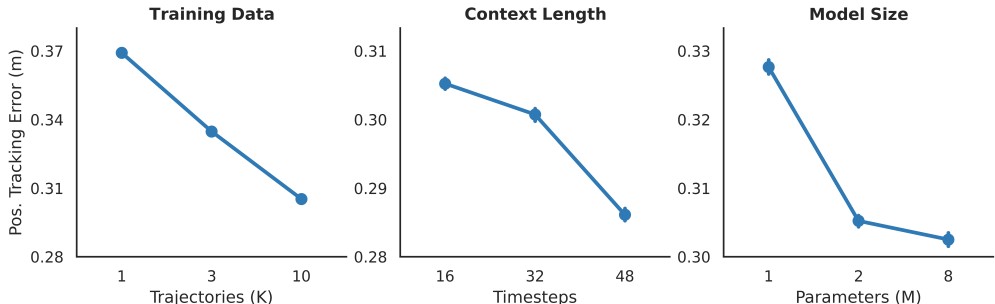

Figure 7: **Scaling studies.** We find that our approach has favorable scaling properties, and that the performance improves with the dataset size (left), context length (middle), and model size (right).

consistently leads to better performance. This suggests that incomplete trajectories can still provide useful signal and is a promising signal for scaling our approach to large datasets of diverse trajectories.

### 5.6 Scaling studies

**Training data.** In Figure 7, left, we report the performance of our model as the size of the training dataset increases. We find that training on more trajectories results in a considerably lower position tracking error. This is a promising signal for scaling our approach to larger datasets in future work.

**Context length.** In Figure 7, middle, we study the effect of increasing the number of tokens used in the context window of the transformer model. We observe that the larger context windows result in better performance which suggests that our model is able to benefit from the additional context.

**Model size.** In Figure 7, right, we compare the models of increasing size (1M, 2M, 8M) by varying the embedding dimension (144, 192, 384), number of attention heads (3, 4, 12), and number of transformer blocks (4, 4, 6). We see that the error decreases monotonically with model size.

### 5.7 Prediction error correlates with performance

We collect 14 models trained with different training recipes, model architectures, data size and types, and compute the tracking and prediction errors for each of them. In Figure 8, we report the tracking and prediction errors of all the models in a single scatter plot. We can see that tracking and prediction error are highly correlated with Pearson coefficient $r = 0.87$. This suggests that the prediction error is predictive of performance, and that models with lower prediction error on the validation set are likely to follow input commands with higher accuracy.

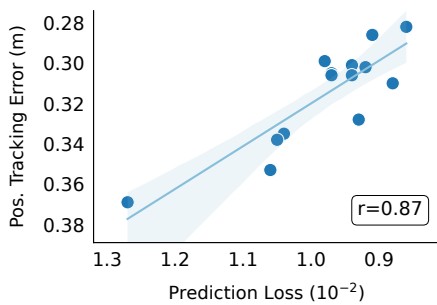

Figure 8: **Prediction error correlation.**

## 6 Discussion

**Limitations.** Our approach still lags behind state-of-the-art MPC controllers and is weaker than the state-of-the-art RL controllers in some regards, most notably robustness. In principle, we could scale our model to millions of YouTube trajectories. However, a major obstacle for doing this is the reliability of computer vision techniques which require a considerable amount of curation in practice.

**Conclusion.** We cast real-world humanoid locomotion as next token prediction. Our model is trained on sensorimotor sequences, which come from a neural network policy, a model-based controller, human motion capture, and videos of humans. We show that our model enables a humanoid robot to walk in the real world zero-shot. These findings suggest a promising path toward learning challenging real-world robotic control tasks by generative modeling of large collections of sensorimotor sequences.

## Contributions

**Ilija Radosavovic** conceptualized and led the project. Generated robot datasets, implemented initial codebase and models, designed and performed experiments, and led the writing of the paper.

**Bike Zhang** worked on retargeting, performed experiments, and contributed to writing.

**Baifeng Shi** designed and implemented improved labeled training, trained and evaluated models, implemented improved unlabeled training, performed experiments, and contributed to writing.

**Jathushan Rajasegaran** designed and implemented improved model architectures, trained and evaluated models, generated human datasets, performed experiments, and contributed to writing.

**Sarthak Kamat** performed experiments and contributed to writing.

**Trevor Darrell** advised BS and SK and provided guidance.

**Koushil Sreenath** advised BZ, provided feedback, and robot equipment.

**Jitendra Malik** advised IR and JR and was the senior advisor to the project.

## Acknowledgements

This work was supported in part by DARPA Machine Common Sense program and ONR MURI program (N00014-21-1-2801) for JM, a DGX A100 donation from NVIDIA, the Hong Kong Centre for Logistics Robotics and The AI Institute for KS, and BAIR's industrial alliance programs. We thank Saner Cakir and Vikas Ummadisetty for help with retargeting, and Alexei Efros for discussions.

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
