# OpenReview forum: "Humanoid Locomotion as Next Token Prediction"
_NeurIPS.cc/2024/Conference — NeurIPS 2024 spotlight_

### Official Review · Reviewer_GH97 · 2024-07-07

**Soundness:** 4
**Presentation:** 4
**Contribution:** 3
**Rating:** 8
**Confidence:** 4

**Summary:**

This paper proposes to use next-token-prediction as a learning objective and train a casual transformer for humanoid locomotion. Compared to previous RL-based methods, the advantage of this method is to fuse data from different sources, including mocap data, videos, RL controller, and MPC controller. The authors successfully show that their model has lower tracking errors when trained with unlabeled data. The trained model is deployed on a Digit humanoid robot and shows robust locomotion across diverse real-world areas.

**Strengths:**

1. This paper introduces the next-token-prediction framework into humanoid robots. The intuition is simple yet very interesting.
2. The usage of different data sources, such as the RL controller and Internet videos, is novel.
3. The real-world experiments are very solid. It is exciting to see such a framework would bring benefits on real-world robot learning.

**Weaknesses:**

1. As shown in Figure 8, the prediction error is correlated to the tracking error. I am curious about how much data the authors need to make these two errors correlated, since from my knowledge, the real world is much more complex and it is usually hard to have such a correlation,
2. Recently, there have been numerous impressive advancements in humanoid robotics. While these developments are exciting, it is unfortunate that most of these works do not open-source their code. I understand the reasons behind keeping the code proprietary, but open-sourcing it would provide significant benefits to the broader community. I believe there are a lot of engineering details behind the simple framework, both in the algorithm and in the real world deployment, which however might not be shown in the paper.

**Questions:**

See weaknesses.

**Limitations:**

The authors have mentioned the limitations.

---

> ### Author Rebuttal · Authors · 2024-08-07
>
> Thank you for your comments and suggestions! Please find our responses below:
>
> > As shown in Figure 8, the prediction error is correlated to the tracking error. I am curious about how much data the authors need to make these two errors correlated, since from my knowledge, the real world is much more complex and it is usually hard to have such a correlation,
>
> We use a total of 30k trajectories for training. Since we evaluate the prediction error on a held out validation set we think that the errors may be correlated in a lower data regime as well (less training data may lead to higher prediction error and higher tracking error as well). Please let us know if this answers your question.
>
> > Recently, there have been numerous impressive advancements in humanoid robotics. While these developments are exciting, it is unfortunate that most of these works do not open-source their code. I understand the reasons behind keeping the code proprietary, but open-sourcing it would provide significant benefits to the broader community. I believe there are a lot of engineering details behind the simple framework, both in the algorithm and in the real world deployment, which however might not be shown in the paper.
>
> Thank you for the suggestion. We commit to open-source the code, models, and data used in this study to facilitate future research in this area.

---

> > ### Comment · Reviewer_GH97 · 2024-08-09
> > **Thank you for the reply**
> >
> > Thank authors for the reply. My questions are mostly addressed.
> >
> > I have raised the score according to the open-source commitment of authors.

---

> > > ### Author Response · Authors · 2024-08-10
> > > **Thank you**
> > >
> > > Thank you for your response and for increasing the score. Please let us know if you have any further questions. We would be happy to address them.

---

### Official Review · Reviewer_qt3e · 2024-07-12

**Soundness:** 4
**Presentation:** 4
**Contribution:** 4
**Rating:** 8
**Confidence:** 4

**Summary:**

Rather than training an RL model to learn how to walk, this paper focuses on an SSL based approach towards humanoid locomotion. Using autoregressive prediction of actions and sensor data, they pre-train and deploy zero shot to the real world. The results are strong, performing better than RL at times.

**Strengths:**

* The idea of using a masked token for unknown modalities is smart and works well
* Generalization to walking backwards is impressive
* The approach is scalable and outperforms RL in many cases

**Weaknesses:**

* There was a lack of details/comprehensiveness on some of the experiments. For example, it was not clear to me how many trials were used for Figure 5. The authors also did not compare to existing MPC and only compared to RL based approaches. Limited details for hparams were given.
* There are more ablations that could be done to further confirm design decisions (see questions).

**Questions:**

* Did the authors do experiments with quantization/VQ? If so it would be nice to see the results for that.
* Ablations for masked token?
* I did not understand this comment: "Rather than predicting the next token in a modality-agnostic way, we make predictions in a modality aligned way. Namely, for each input token we predict the next token of the same modality."
* Ablations on which way to pre-train--noisy and incomplete or joint?
* Not much information on the modalities themselves/examples modalities. If there are tons of motors/sensory information used at any given point are all of these predicted at once, with the same model? In that case, how is the causal mask constructed?
* What is the prediction error in Table 1c?
* Was predicting both \hat{o}_{i+1} and \hat{a}_{i+1} at the same time tried?

**Limitations:**

Yes

---

> ### Author Rebuttal · Authors · 2024-08-07
>
> Thank you for your comments and suggestions! Please find our responses below:
>
> **Additional details on some experiments.**
>
> Our goal in Figure 5 is to give a qualitative sense and we use one trial for each command. The following table shows a quantitative comparison between our model, RL, and MPC models. We report the mean velocity tracking error of different models under different yaw commands (250 different yaw values sampled in total).
>
> |             |           |           |           |           |           |
> | ----------- | --------- | --------- | --------- | --------- | --------- |
> | Yaw (rad/s) | 0.0       | 0.1       | 0.2       | 0.3       | 0.4       |
> | MPC         | 0.108     | 0.112     | 0.112     | 0.123     | 0.119     |
> | RL          | 0.082     | 0.083     | 0.090     | 0.105     | 0.110     |
> | Ours        | **0.070** | **0.077** | **0.078** | **0.081** | **0.089** |
>
> We can see that MPC has the highest tracking error among the three methods while our model has the better performance across all the commands. We will update the comparisons to MPC, as well as the hyperparameters, in the next version of our paper.
>
> **Ablation on quantization.**
>
> Our model uses MSE loss on continuous observations and actions. We compare it to using cross entropy (CE) loss on quantized observations and actions. We use uniform binning for quantization, i.e., for each dimension of the observations and actions, we split the value of that dimension into N uniform bins and the observation/action prediction is formulated as a classification problem of predicting which bin the value of that dimension falls into. We use N=100 for all dimensions. The results are shown below:
>
> |                  |                       |                     |
> | ---------------- | --------------------- | ------------------- |
> |                  | Continuous (MSE loss) | Quantized (CE loss) |
> | Prediction Error | **1.39**              | 10.41               |
>
> We can see that using MSE loss has a better performance. Note that here we only use a vanilla quantization method of uniform binning and using more advanced approaches such as non-uniform binning or vector quantization might provide better performance.
>
> **Ablation on mask token.**
>
> Our model uses mask token to replace the missing modalities in model-based, MoCap, and human video data. We compare using a leanable mask token with using a constant vector (e.g., zero vector) to fill in the missing modalities. Results are shown below:
>
> |                  |                     |                 |
> | ---------------- | ------------------- | --------------- |
> |                  | Learnable mask token | Constant vector |
> | Prediction Error | **1.39**                | 1.42            |
>
> We can see that using mask token is slightly better than using a constant vector.
>
> **Clarification on modality-aligned prediction.**
>
> Given a sequence of interleaved observation and action tokens [$o_1$, $a_1$, …, $o_{t-1}$, $a_{t-1}$, $o_t$, $a_t$, …], the naive next token prediction will predict $o_t$ from $a_{t-1}$ and then $a_t$ from $o_t$. For modality-aligned prediction, we predict $o_t$ from $o_{t-1}$ and $a_t$ from $a_{t-1}$, i.e., we always predict the token from the previous token that belongs to the same modality.
>
> **Ablation on noisy pre-training vs. joint training.**
>
> We compare two ways of training our model: 1) first pre-training on noisy observation prediction on MPC, MoCap, and human video data, and then fine-tuning on action prediction on RL data. 2) training the model jointly on observation and action prediction on all data. Results are shown below:
>
> |                |                    |                |
> | -------------- | ------------------ | -------------- |
> |                | Noisy pre-training | Joint training |
> | Tracking Error | 0.311              | **0.310**      |
>
> We can see that joint training is slightly better. The results are also reported in Table 1c in the paper.
>
> **Details on the modalities used.**
>
> There are in total two modalities: observation and action. The observation is a 102-dim vector which is the concatenation of the joint positions of each joint/motor, the velocities of each joint, the velocity of the body, the gravity, the command, etc. The action is a 36-dim vector which contains the commands it will send to each motor for the next step. During inference, at each step the model predicts both the observation and action vector for this step. Assuming the input is the history of observations and actions [$o_1$, $a_1$, …, $o_{t-1}$, $a_{t-1}$], the model predicts $o_t$ and $a_t$ at the t-th step. The causal mask is constructed such as any modality in each step (e.g., $o_{t-1}$) will only see other modalities in the same step ($a_{t-1}$) as well as all the modalities in the previous steps (e.g., $o_{t-2}$ and $a_{t-2}$).
>
> **The prediction error for stage training (Table 1c).**
>
> For stage training, the model is pre-trained to predict observations in the first stage, and then fine-tuned with action prediction in the second stage. Since the prediction error counts in both observation prediction and action prediction, and the model is not supervised with observation prediction in the second stage, it is probably not fair to compare with the model that is trained jointly on observation and action prediction, which is why we didn’t report the number. For reference, the prediction error for stage pretraining is 42.22 which is way higher than the prediction error of joint training (0.88).
>
> **Was predicting both observation and action at the same time tried?**
>
> Our model predicts both observation and action at the same time. Please also see the clarification on modality-aligned prediction above.

---

> > ### Author Response · Authors · 2024-08-12
> > **Author response summary**
> >
> > We appreciate your positive review and helpful suggestions. We wanted to provide a summary of the additional information provided in our response for your reference.
> >
> > Additional experiments on:
> > - MPC comparisons - ours > RL > MPC
> > - Quantization ablation - continuous > discrete
> > - Mask token ablation - learnable > constant
> > - Pre-training ablation - joint > staged
> >
> > Additional details on:
> > - Modality prediction - predict next token of the same modality
> > - Modalities used - observations and actions
> > - Prediction error - provided for reference
> > - Simultaneous prediction - used in our models
> >
> > Thank you again for your valuable feedback. We would be happy to provide any further information if needed.

---

> > > ### Comment · Reviewer_qt3e · 2024-08-12
> > >
> > > Thank you for addressing my comments, I am happy with the author’s rebuttal.
> > >
> > > However, the lack of comprehensiveness in discussing related works is slightly concerning. I was not knowledgeable about the work described by reviewer FS4P, which makes the contribution of this work less meritous. I believe the addition of an “Autoregressive pre-training for robotics” header in the related works section would be appropriate. Thus, I am lowering my score slightly.
> > >
> > > It’s worth noting that I still believe the work has a lot of merit, such as through masking actions and through applying autoregressive pre-training to humanoid locomotion. The results were also very impressive. I also agree with other reviewers that the eventual open source release is essential for this work. Thanks for addressing my concerns!

---

> ### Author Response · Authors · 2024-08-13
> **Response by Authors**
>
> Thank you for your response. We are glad to hear that we addressed all of your initial comments. We would like to offer some clarifications regarding the related work discussion.
>
> We believe that reviewer FS4P’s suggestion about an additional reference in robotic manipulation was a minor point, which we agreed to include. Reviewer FS4P acknowledged that this concern was fully addressed in their response and also noted the novelty of our work in their review.
>
> We want to emphasize that we do not claim "Autoregressive Pre-training for Robotics" as our primary contribution. In fact, we already have the "Transformers for Robotics" section in the related work that discusses a number of papers on autoregressive pre-training in robotics. We are happy to expand this section further.
>
> The novelty of our work lies in casting real-world humanoid locomotion as a joint distribution learning problem with transformers. We further show that this formulation enables training with noisy or missing data (e.g., derived from videos) and achieves excellent results in the real world.
>
> While our work can have excellent impact on multiple areas, the paper and title focus on humanoid locomotion. We chose this problem as a test bed because it is particularly challenging due to its highly dynamic, complex nature. The success of our method in this domain is particularly noteworthy (few prior approaches used learning and the ones that did rely on RL).
>
> Given these clarifications and the original review, we believe the original score more accurately reflects our contribution and that it would be unfair to reduce it. We respectfully ask you to reconsider maintaining your initial score.

---

> > ### Comment · Reviewer_qt3e · 2024-08-13
> >
> > I appreciate the response. Perhaps, this was a mistake on my end, as while reading the paper I believed that the idea of autoregressive pre-training for robotics was a primary contribution. Adding to this, I think that the lack of inclusion of autoregressive pre-training approaches in the "Transformer for Robotics" section is of slight concern, as *autoregressive pre-training approaches for robotics are the most similar to this work.* (I say lack of inclusion because all approaches listed were either not pre-training or not autoregressive as far as I could tell). Together, these enhanced my initial beliefs regarding the merit of this work.
> >
> > I acknowledge the contributions discussed and the excellent results. Hence, I think a score of 8 is fitting. This updated score is based off of the planned inclusion of more autoregressive pre-training approaches in the related works.

---

> ### Author Response · Authors · 2024-08-13
> **Response by Authors**
>
> Thank you for your response. We appreciate your acknowledgement of our contributions and results. We would like to clarify a few points below:
>
> - Autoregressive pre-training is not a primary contribution of our work. While our approach can be used with pre-training, our method does not involve pre-training. We are "just" training a model from scratch and deploying it zero shot, without multi-stage training or fine-tuning.
>
> - We are more than willing to expand the “Transformers for Robotics” section with work on autoregressive pre-training in robotics to provide a more comprehensive context.
>
> - Our primary contribution lies in formulating real-world humanoid locomotion as a joint distribution learning problem, enabling training with noisy or missing data and achieving very strong results in a particularly challenging domain.
>
> Thank you for your consideration.

---

### Official Review · Reviewer_FS4P · 2024-07-12

**Soundness:** 3
**Presentation:** 3
**Contribution:** 3
**Rating:** 7
**Confidence:** 4

**Summary:**

This paper trained an autoregressive transformer for humanoid robot walking control using four types of data. The data sources included trajectories rolled out by RL Policy and Scripted Method, existing datasets, and human poses from YouTube videos. This work successfully enabled the humanoid robot to walk in urban environments without falling. The study explored how to train using data with missing actions, specifically by utilizing observation data and observation-action data and demonstrated the effectiveness of training with missing data. The robot in this work does not have visual capabilities.

**Strengths:**

1. The research problem in this work is very novel, and it has been verified on real humanoid. The effects shown in the video are excellent.
2. This work is dedicated to applying autoregressive Transformers to humanoid locomotion, which is a promising direction.
3. This work explores the usefulness of training with action-free data.

**Weaknesses:**

1. Claiming that training with video might be overclaiming; in fact, it only uses human poses from the video, which is quite different from training with video.
2. Pre-training with action-free data is not a novel contribution.
3. Based on the model architecture and training data, it appears that the robot can only walk randomly without controlling speed or turning. However, in Figure 5, it seems to control the robot’s walking. How is this achieved? What exactly is the observation? It doesn't seem to be the observed images. How is the observation transformed into R^m?
4. For the four types of data collected—Neural Net Controller, Model-based Controller, MoCap, Internet Videos—the authors did not conduct ablation experiments to verify the usefulness of each type of data. Especially the MoCap and Internet Videos from inverse dynamics are highly questionable in their usefulness. Considering that the authors did not prove the usefulness of data from inverse dynamics, the paper would degrade to merely fitting some existing policies with a transformer, significantly limiting its practical value.
5. The paper does not specify the control frequency or how the control frequency changes with model size, nor does it explain how to deploy the robot in the field. Where is the GPU located?
6. The pre-training method used in GR1[1] is very similar to this work. The only difference is that GR1 uses two-stage training while this paper uses joint training. Both output observation-action, but this paper does not discuss this.

[1] UNLEASHING LARGE-SCALE VIDEO GENERATIVE PRE-TRAINING FOR VISUAL ROBOT MANIPULATION

Writing suggestions:
1. Figure 3 is currently unclear. I suggest adding sequence numbers to the tokens to distinguish their order and using superscripts to differentiate predicted tokens from given tokens.
2. Equations 5-10 are not very necessary and could be condensed into 1-2 lines. More space could be used to explain the training data for the transformer, which is more important.
3. Using subplots to redraw Figures 6 and 7 would be clearer.

**Questions:**

1. How to control the robot's walking?
2. What exactly is Observation, and how is it transformed into R^m?
3. From my understanding, the Neural Net Controller is the RL policy, the Model-based Controller is a scripted policy (non-learning method), and MoCap is a robot dataset, but not for the used Agility Robotics. Is my understanding correct?
4. Is it possible to conduct ablation experiments on each type of data?
5. What is the control frequency, what GPU and model size are used during deployment, and is the GPU placed on the robot or on a nearby PC?

**Limitations:**

1. One key limitation of the article is the lack of visual input, which is not discussed in the paper.
2. Another limitation of the article is that it does not show how the real humanoid turns; it appears that the robot in the video only walks randomly.
3. Is it compliant to use YouTube videos, especially those featuring humans as the main subject?

---

> ### Author Rebuttal · Authors · 2024-08-07
>
> Thank you for your comments and suggestions! Please find our responses below:
>
> > Claiming that training with video might be overclaiming; in fact, it only uses human poses from the video, which is quite different from training with video.
>
> We pre-process human videos using a pre-trained transformer model to extract human poses. We will clarify in the revised manuscript.
>
> > Pre-training with action-free data is not a novel contribution.
>
> We did not intend to claim that training with action-free data is unique to our work. Indeed, there has been prior work that studied this problem (we will include a discussion in the revised version of the manuscript). In this work, we show that a benefit of our approach is that it enables us to leverage action-free data for humanoid locomotion.
>
> > it appears that the robot can only walk randomly without controlling speed or turning. However, in Figure 5, it seems to control the robot’s walking. How is this achieved?
>
> Our controller supports omni-directional walking at varying speeds of [-1,1] m/s. We will include additional videos in the revised supplementary materials. The walking speed and direction are specified as a 3-dim vector containing the desired linear velocity on x-axis and y-axis, and the desired angular velocity around the z axis. We specify the command as a part of the observation vector. For real-world deployment, we vary the command in real time via a joystick.
>
> > What exactly is the observation? It doesn't seem to be the observed images. How is the observation transformed into R^m?
>
> The observation is a 102-dim vector which is the concatenation of the joint positions of each joint (26-dim), the velocities of each joint (26-dim), the linear and angular velocity of the body (6-dim), the gravity (3-dim), the clock input (2-dim), the command (3-dim) and the previous action containing the commanded position of each joint (20-dim) and the updated P and D gains for the PD controller (16-dim). The robot is blind and does not take visual information as input. Observations are projected using a single linear layer into the embedding dimension.
>
> We will include the additional details in the revised version of the manuscript.
>
> > the authors did not conduct ablation experiments to verify the usefulness of each type of data.
>
> Following reviewer suggestion, we perform the ablations below:
>
> |          |            |
> | -------- | ---------- |
> | Data     | Pred. Err. |
> | NN       | 4.17       |
> | NN+MPC   | 2.83       |
> | NN+MoCap | 2.66       |
> | NN+Video | 3.19       |
>
> |                    |            |
> | ------------------ | ---------- |
> | Data               | Pred. Err. |
> | NN                 | 4.17       |
> | NN+MPC             | 2.83       |
> | NN+MPC+MoCap       | 2.28       |
> | NN+MPC+MoCap+Video | 2.23       |
>
> We see that each of the data sources leads to gains in performance individually and in aggregate.
>
> > The paper does not specify the control frequency or how the control frequency changes with model size, nor does it explain how to deploy the robot in the field. Where is the GPU located?
>
> For all our experiments and all the model sizes we try in this work (1M, 2M, and 8M), the model is run at 50Hz during inference. The robot does not have a GPU on board and the model is run on the CPU of the on-board Intel NUC computer. In all of our outdoor experiments, everything is running on board. We use the 2M model by default for all experiments.
>
> > Related Work (GR1).
>
> Thank you for the reference. While we focus on a different task (humanoid locomotion vs. manipulation) and use a different training strategy (staged training vs. joint training), there are some shared observations such as predicting observation can help action prediction. We will include the discussion in the revised version of the paper.
>
> > Writing and formatting.
>
> Thanks for the suggestions. We will incorporate.
>
> > How to control the robot's walking?
> > What exactly is Observation, and how is it transformed into R^m?
>
> Please see our comments above.
>
> > From my understanding, the Neural Net Controller is the RL policy, the Model-based Controller is a scripted policy (non-learning method), and MoCap is a robot dataset, but not for the used Agility Robotics. Is my understanding correct?
>
> Neural network controller is an RL policy from [33]. Please note that the Model-based Controller is a state-of-the art classical controller from Agility Robotics (for a dynamic task like humanoid locomotion it would be hard to script a policy). MoCap is not a robot dataset but a dataset of humans from [28, 24] (we retarget the human pose trajectories to humanoid locomotion trajectories via inverse kinematics).
>
> > One key limitation of the article is the lack of visual input, which is not discussed in the paper.
>
> We focus on blind locomotion in this work which is a challenging problem. We intend to extend our approach to vision in future work. We will include a discussion in the revised manuscript.
>
> > Another limitation of the article is that it does not show how the real humanoid turns;
>
> Our controller supports omni-directional walking and can turn in any direction. Please see above for additional discussion. We will include videos in the updated version.
>
> > Is it compliant to use YouTube videos, especially those featuring humans as the main subject?
>
> The videos we use are a subset of public videos from research datasets (Kinetics, PoseTrack) and we pre-process the videos to extract poses which anonymizes the individuals in the videos.

---

> > ### Comment · Reviewer_FS4P · 2024-08-09
> >
> > Thank you very much for your response. I appreciate the effort in addressing my questions, and most of them have been satisfactorily resolved. However, I have some concerns regarding the reproducibility of the paper. It seems challenging to obtain the four datasets mentioned in the paper based solely on the descriptions provided. While I believe the paper offers valuable insights, the limited details, absence of detailed information in the appendix, and lack of accompanying code might affect its overall impact on the community. Considering these factors, I believe it may be best to maintain my previous score. Thank you once again for your thoughtful response.

---

> > > ### Author Response · Authors · 2024-08-09
> > > **Response by Authors**
> > >
> > > Thank you for your response. Please find our responses below:
> > >
> > > > I appreciate the effort in addressing my questions, and most of them have been satisfactorily resolved
> > >
> > > We are glad to hear you found our responses satisfactory. To ensure we have fully addressed all of your concerns, please let us know if there are any specific questions or points that you feel remain unresolved or require further clarification. We would be happy to address them.
> > >
> > > > However, I have some concerns regarding the reproducibility of the paper. It seems challenging to obtain the four datasets mentioned in the paper based solely on the descriptions provided. While I believe the paper offers valuable insights, the limited details, absence of detailed information in the appendix, and lack of accompanying code might affect its overall impact on the community.
> > >
> > > We commit to release all of the materials to fully reproduce the paper, including the four datasets, accompanying code, and detailed documentation. We will further include detailed descriptions of the datasets in the appendix, covering data collection methods, pre-processing steps, and any other relevant details. Please let us know if there is any additional information that you found lacking, and we would be happy to include it in the revised manuscript.

---

> > > > ### Comment · Reviewer_FS4P · 2024-08-09
> > > >
> > > > Thank you for your response.
> > > > My remaining concerns mainly focus on the actual capabilities of this robot:
> > > > The project page doesn't include videos of the robot turning; can it actually turn in the real world or walk for an extended period of time?
> > > > However, the project page seems to be inaccessible now. It would be interesting to see its obstacle avoidance capabilities under control.
> > > > Additionally, if the robot needs to go uphill or downhill slightly, would it fail? And it will be fine to analyze its robustness.
> > > > If this method were extended to enable humanoid robots to go up and down stairs, what challenges would arise?
> > > >
> > > > Some of the suggestions may go beyond the scope of this paper, but considering the authors’ commitment to open-source, I have raised my score.

---

> ### Author Response · Authors · 2024-08-10
> **Response by Authors**
>
> Thank you for your response and for raising the score. Please find our responses below:
>
> > The project page doesn't include videos of the robot turning; can it actually turn in the real world or walk for an extended period of time? However, the project page seems to be inaccessible now. It would be interesting to see its obstacle avoidance capabilities under control.
>
> Our policy can turn in all directions and walk for extended periods of time, up to 2 hours on a single battery charge. We have extensively tested these capabilities during a week-long deployment in San Francisco. Thanks to the responsive and accurate command following, our robot can avoid obstacles and navigate busy city streets with pedestrians.
>
> As an additional data point, our omnidirectional walking capabilities are comparable to the state-of-the-art RL approach in Figure 3 of [33], with improved command following as demonstrated in Figures 5 and Figure 6 (left) of our manuscript.
>
> We apologize for the inaccessible project page. It seems that our GitHub-hosted anonymous website was suspended by GitHub, which we are working to resolve with GitHub. Per the rebuttal instructions, it seems that we are not allowed to provide a new link to external pages in the rebuttal. We welcome any suggestions on how we might share the videos demonstrating these capabilities within the constraints of the rebuttal process and commit to include them in the release.
>
> > Additionally, if the robot needs to go uphill or downhill slightly, would it fail? And it will be fine to analyze its robustness.
>
> Our policy can handle gentle slopes, and we have tested it in the real world on inclines of up to 8.7% (5 degrees). For example, Figure 1 (row 3, column 1) shows the robot walking up a 7% slope in San Francisco.
>
> Regarding robustness, our approach trained purely on offline trajectories without any domain randomization has shown surprising effectiveness in walking in the real world. However, as we acknowledge in the limitations section, our policies may be less robust to sudden, large disturbances compared to RL policies.
>
> In follow-up experiments, we found that pre-training policies with our proposed framework and fine-tuning them with a small amount of RL leads to excellent results, enabling rapid acquisition of new capabilities like walking on steeper slopes and robustness to large disturbances.
>
> > If this method were extended to enable humanoid robots to go up and down stairs, what challenges would arise?
>
> To extend our method to enable the robot to go up and down stairs, we would need to incorporate vision to guide the foot placement. Our method can be readily extended to achieve this by incorporating images (potentially pre-processed using a vision encoder) as part of observations in addition to proprioception.
>
> One challenge would be in collecting training trajectories that contain visual inputs. This would be relatively straightforward for trajectories collected using the same robot body (e.g., via prior RL or MPC controllers). To incorporate human trajectories from mocap or videos, we would need to include ego vision inputs where available (e.g., first-person videos) and use our strategy for training with missing data for trajectories without ego vision (e.g., third-person videos).
>
> > Some of the suggestions may go beyond the scope of this paper
>
> Thank you for raising these questions. We will include the discussion of capabilities, limitations, and future work in the revised version of the manuscript with an extended discussion section.

---

> > ### Author Response · Authors · 2024-08-11
> > **Added videos of the robot turning to the project page**
> >
> > We have resolved the hosting issue with GitHub and our anonymous project page is accessible again via the link provided in the original manuscript.
> >
> > As requested, we have included videos of the robot turning in the real world under the “Turning in the Real World” section at the bottom of the page.
> >
> > Please let us know if all your concerns are now fully addressed. We would be happy to provide any additional information.

---

> > > ### Comment · Reviewer_FS4P · 2024-08-13
> > >
> > > Thank you for your response. I appreciate your excellent experiments; it's fascinating to see how they can be applied to real-world robotics.
> > >
> > > I agree with qt3e's point—a more detailed autoregressive model for the robotics section would better reflect the contributions of this paper.

---

> ### Author Response · Authors · 2024-08-13
> **Response by Authors**
>
> Thank you for your response. We are glad to hear that we have addressed your remaining concerns regarding robot capabilities and appreciate your acknowledgement of our results.
>
> We are more than willing to include a more detailed autoregressive models for robotics section. We will incorporate the suggestion in the revised manuscript.
>
> We believe that we have addressed all of your concerns from the initial review and subsequent discussions. Given these and in light of the strengths you and all other reviewers have highlighted, we respectfully ask you to consider increasing the overall score to better reflect our contributions.

---

> > ### Comment · Reviewer_FS4P · 2024-08-14
> >
> > I believe the score I have given is appropriate. This paper represents yet another success of Next Token Prediction in a different domain, a success that has been replicated many times. There are few new techniques introduced. Essentially, papers in this paradigm primarily involve preparing data, training, and testing, without much technical complexity. I also took into account that this paper does not achieve significant breakthroughs; previous methods could also achieve tasks such as walking and turning, just not using next token prediction.
> >
> > A notable drawback is that the scenarios considered are particularly simple, with only 100 dimensions of the robot's body state being considered as observations. The challenge might lie in processing and unifying large amounts of data. The primary reason this paper achieves such a high score maybe is that it effectively addresses the problem of studying humanoid locomotion using AI methods and works well in the real world. If the same approach were applied to robotic manipulation, it might not score as highly. However, the manipulation problem involves more complex considerations than humanoid locomotion, such as language instructions, visual perception, and the robot's state.
> > Despite some limitations, I still believe this work is valuable because it demonstrates that next token prediction works well in such high degrees of freedom, given the appropriate data.
> > Considering the author's commitment to open-sourcing their work, which could significantly advance the progress of humanoid locomotion—a field that has been relatively unexplored—I believe a score of 7 is reasonable.

---

### Official Review · Reviewer_rJUi · 2024-07-22

**Soundness:** 4
**Presentation:** 4
**Contribution:** 3
**Rating:** 8
**Confidence:** 4

**Summary:**

This work view the robot locomotion control problem as an next token prediction problem. A causal transformer is trained autoregressively on various sources of data. The performance on a full-sized humanoid robot's locomotion indicates that this formulation can be a promising path for complex robotic control problems to incorporate diverse sources of data.

**Strengths:**

1. This work presents a promising way to scale robotic learning in terms of data via generative modeling.
2. Real-world experiments, and some metrics indicate the comparable performance of the proposed method to RL-based / model-based controller.

**Weaknesses:**

1. In Sec. 3.6 Model inference, the first step might not follow the statement.
> At inference time, our transformer model will always have access to observation-action pairs.

2. It lacks description on the inference frequency, considering the large scale pretraining with real-world locomotion task.

3. It might be beneficial to ablate on how different sources of the data affect the performance.

**Questions:**

1. In Table 1(c), why no  Pred. Err. for staged training?

2. In Sec. 4.3, can the authors detail the inverse kinematics problem they solve?
> In order to use these trajectories for training a robot, we solve an inverse kinematics problem to
183 find the corresponding robot poses.

3. In Sec. 4.4, can the authors detail on the filter strategy?
> Once we retarget the motion from the Internet videos to humanoid trajectories, we filter the trajectories with the low optimization cost.

4. How many Internet data are used? This seems not stated in the paper.

**Limitations:**

As the authors noted, data from human video come with the cost of being noisy as a result of our progress on the computer vision techniques. It would be helpful to elaborate on current results of using Internet videos.

---

> ### Author Rebuttal · Authors · 2024-08-07
>
> Thank you for the comments and suggestions! Please find our responses below:
>
> > In Sec. 3.6 Model inference, the first step might not follow the statement.
>
> Thanks for pointing this out. Indeed, at the first step we have the current observation but not the action from the previous step. We use zero padding to fill in the missing actions for the initial step in both training and inference. We will clarify this in the next version of the paper.
>
> > It lacks description on the inference frequency
>
> During inference, our neural network model predicts desired joint positions as well as PD gains at 50 Hz. These are used as targets for a low-level PD controller that runs at 2000 Hz.
>
> > It might be beneficial to ablate on how different sources of the data affect the performance
>
> Thank you for the suggestion. We perform detailed ablations below:
>
> |          |            |
> | -------- | ---------- |
> | Data     | Pred. Err. |
> | NN       | 4.17       |
> | NN+MPC   | 2.83       |
> | NN+MoCap | 2.66       |
> | NN+Video | 3.19       |
>
> |                    |            |
> | ------------------ | ---------- |
> | Data               | Pred. Err. |
> | NN                 | 4.17       |
> | NN+MPC             | 2.83       |
> | NN+MPC+MoCap       | 2.28       |
> | NN+MPC+MoCap+Video | 2.23       |
>
> We find that each of the data sources leads to gains in performance individually (first table) and in aggregate (second table).
>
> > In Table 1(c), why no Pred. Err. for staged training?
>
> For stage training, the model is pre-trained to predict observations in the first stage, and then fine-tuned with action prediction in the second stage. Since the prediction error counts in both observation prediction and action prediction, and the model is not supervised with observation prediction in the second stage, it is probably not fair to compare with the model that is trained jointly on observation and action prediction. For reference, the prediction error for stage pre-training is 42.22 which is considerably higher than the prediction error of joint training (0.88).
>
> > In Sec. 4.3, can the authors detail the inverse kinematics problem they solve?
>
> We formulate an inverse kinematics optimization problem as follows:
>
> $$
> \begin{align}
>     \min_{\substack{\mathbf{q}[t], \mathbf{\dot{q}}[t]}} ~ & \sum_{t=1}^{N} \varphi^{\text{traj}}[t] + \varphi^{\text{reg}}[t] \\
>     \text{s.t.} ~
>     & \mathbf{q}[t+1] = \mathbf{q}[t] + \frac{\mathbf{\dot{q}}[t+1] + \mathbf{\dot{q}}[t]}{2} dt, \\
>     & \mathbf{q} \in \mathcal{Q}, \mathbf{\dot{q}} \in \mathcal{V}
> \end{align}
> $$
>
> where $\mathbf{q}$ is the robot state in the generalized coordinates, and $N$ and $dt$ are the optimization horizon and sampling time. The optimization variables include $\mathbf{q}$, $\mathbf{\dot{q}}$.  For constraints, we include the Euler integration of posture $\mathbf{q}$, constrain the range of $\mathbf{q}$ and $\mathbf{\dot{q}}$ to their admissible sets $\mathcal{Q}$ and $\mathcal{V}$. In the cost function, $\varphi^{\text{traj}}$ tracks keypoint locations from human trajectories, and $\varphi^{\text{reg}}$ represents the regularization costs that include joint velocity minimization and smoothness.
>
> > In Sec. 4.4, can the authors detail on the filter strategy?
>
> We empirically set a threshold on the reconstruction loss of the inverse kinematics retargeting and filter out the trajectories below the threshold.
>
> > How many Internet data are used? This seems not stated in the paper.
>
> We use 1k human videos, which is the same number of trajectories we use for MoCap data.
>
> > It would be helpful to elaborate on current results of using Internet videos.
>
> Please see the additional ablations reported above. We believe that this is a promising signal for using video data.

---

> > ### Author Response · Authors · 2024-08-12
> > **Kind reminder for feedback - 2 days left for discussion**
> >
> > Thank you again for your time and effort spent on providing a careful review of our paper.
> >
> > We performed new experiments on how different data sources affect performance. We find that (1) each of the data sources improves the performance individually and (2) that different data sources are complementary and lead to greater gains in aggregate. These detailed ablations demonstrate that our approach can benefit from different data sources.
> >
> > We also provided additional details on:
> > - model inference
> > - inference frequency
> > - prediction errors
> > - inverse kinematics
> > - filtering strategy
> > - number of videos
> >
> > To ensure we have fully addressed all of your concerns, please let us know if there are any specific questions or points that you feel remain unresolved or require further clarification. We would be happy to address them.
> >
> > Many thanks

---

> ### Comment · Reviewer_rJUi · 2024-08-13
> **Thank you for the rebuttal.**
>
> Thank authors for the detailed response. My questions are mostly addressed. My concern left is that will the largest model work at 50Hz, i.e. whether the increased model size with generative pretraining, hinders the inference. It would be great to discuss the minimum required frequency for functioning, and its relation to model size.
>
> I have raised the score according to the detailed open-source commitment of authors.

---

> ### Author Response · Authors · 2024-08-14
> **Thank you**
>
> Thank you for your response and for increasing the score. We think that the challenge of balancing model size and inference speed in real-time settings like ours is very interesting. Please find some comments below.
>
> The minimal frequency for functioning is determined by the highly dynamic nature of the problem (unstable bipedal robot with a big upper body with a lot of mass and inertia). We might be able to reduce it a bit (e.g., 40Hz) but probably not much. Our current models are relatively small (up to 8M parameters) and can run within 50Hz on the on-board CPU computer.
>
> We think that it is likely that the improvements in inference software (e.g., low precision inference), on-board compute (e.g., access to a GPU), training recipes (e.g., distillation), and modeling (e.g., sparse architectures) will offset these challenges and enable us to keep using larger models within the inference speed constraints.
>
> We find the modeling angle to be a particularly interesting direction for future work. For example, one could imagine architectures with more parameters than flops, where different parameters are activated at different frequencies and different parts of the model are executed asynchronously.
>
> We will include the additional discussion and an analysis of the model size and inference speed in the revised manuscript. Thank you for the suggestions.

---

### Decision · Program_Chairs · 2024-09-25

**Decision:**

Accept (spotlight)

**Comment:**

This paper presents a novel approach to humanoid locomotion control by framing it as a next token prediction problem using a causal transformer model. The authors train their model on a diverse dataset including simulated trajectories, motion capture data, and human poses extracted from YouTube videos. Trained on this diverse collection of data, the model achieves zero-shot transfer to a real humanoid robot, enabling successful locomotion in complex urban environments.

The reviewers were generally very positive about the paper. They highlighted several key strengths: the novelty of formulating a complex humanoid locomotion problem as a next token prediction problem, the ability to leverage diverse data sources including action-free data like human videos, strong real-world results demonstrating zero-shot transfer to a physical robot, performance that outpaces reinforcement learning baselines in some scenarios, and generalization to unseen behaviors like walking backwards. These contributions represent a significant advancement for humanoids, and essentially validate the effectiveness of scaling to this locomotion problem.

However, the reviewers also identified some weaknesses. These included limited ablation studies and analysis of different data sources, lack of clarity around some experimental details and hyperparameters. The authors provided thorough responses to these concerns in their rebuttal, including additional ablation results and clarifications on model details. While there are some slight limitations in terms of analysis depth, these do not significantly detract from the overall contribution. There were also some slight concerns highlighted in the context that while posing humanoid locomotion as a next token prediction problem constitutes a very interesting application, it is objectively less complex than a domain like manipulation which is held to a higher standard, where it's expected to combine several modalities such as language, multiple sensors etc. But I do think this paper demonstrates a strong pipeline in its own way in a relatively new area, specifically through the fact that the dataset was constructed from diverse sources.

To improve the final version, the authors should expand the related work section, particularly highlighting recent works on autoregressive pre-training for robotics (which has been tackled quite a bit in recent times in varied domains), include the additional ablation results provided in the rebuttal, clarify experimental details and hyperparameters.

In conclusion, this paper represents a good contribution to the field of humanoid robotics and is well-suited for presentation at NeurIPS. Its novel formulation of humanoid locomotion control and strong real-world results have the potential to spark new research directions in the field. The authors' commitment to open-sourcing their work further enhances its potential impact on the research community, and I would highly recommend the authors do so.